# TSMixer: An All-MLP Architecture for Time Series Forecasting

**Si-An Chen**                      *d09922007@ntu.edu.tw*
*National Taiwan University*
*Google Cloud AI Research*

**Chun-Liang Li**                      *chunliang@google.com*
*Google Cloud AI Research*

**Nathanael C. Yoder**                    *nyoder@google.com*
*Google Cloud AI Research*

**Sercan Ö. Arık**                      *soarik@google.com*
*Google Cloud AI Research*

**Tomas Pfister**                      *tpfister@google.com*
*Google Cloud AI Research*

**Reviewed on OpenReview:** *https://openreview.net/forum?id=wbpxTuXgm0*

## Abstract

Real-world time-series datasets are often multivariate with complex dynamics. To capture this complexity, high capacity architectures like recurrent- or attention-based sequential deep learning models have become popular. However, recent work demonstrates that simple univariate linear models can outperform such deep learning models on several commonly used academic benchmarks. Extending them, in this paper, we investigate the capabilities of linear models for time-series forecasting and present Time-Series Mixer (TSMixer), a novel architecture designed by stacking multi-layer perceptrons (MLPs). TSMixer is based on mixing operations along both the time and feature dimensions to extract information efficiently. On popular academic benchmarks, the simple-to-implement TSMixer is comparable to specialized state-of-the-art models that leverage the inductive biases of specific benchmarks. On the challenging and large scale M5 benchmark, a real-world retail dataset, TSMixer demonstrates superior performance compared to the state-of-the-art alternatives. Our results underline the importance of efficiently utilizing cross-variate and auxiliary information for improving the performance of time series forecasting. We present various analyses to shed light into the capabilities of TSMixer. The design paradigms utilized in TSMixer are expected to open new horizons for deep learning-based time series forecasting. The implementation is available at: `https://github.com/google-research/google-research/tree/master/tsmixer`.

## 1 Introduction

Time series forecasting is a prevalent problem in numerous real-world use cases, such as for forecasting of demand of products (Böse et al., 2017; Courty & Li, 1999), pandemic spread (Zhang & Nawata, 2018), and inflation rates (Capistrán et al., 2010). The forecastability of time series data often originates from three major aspects:

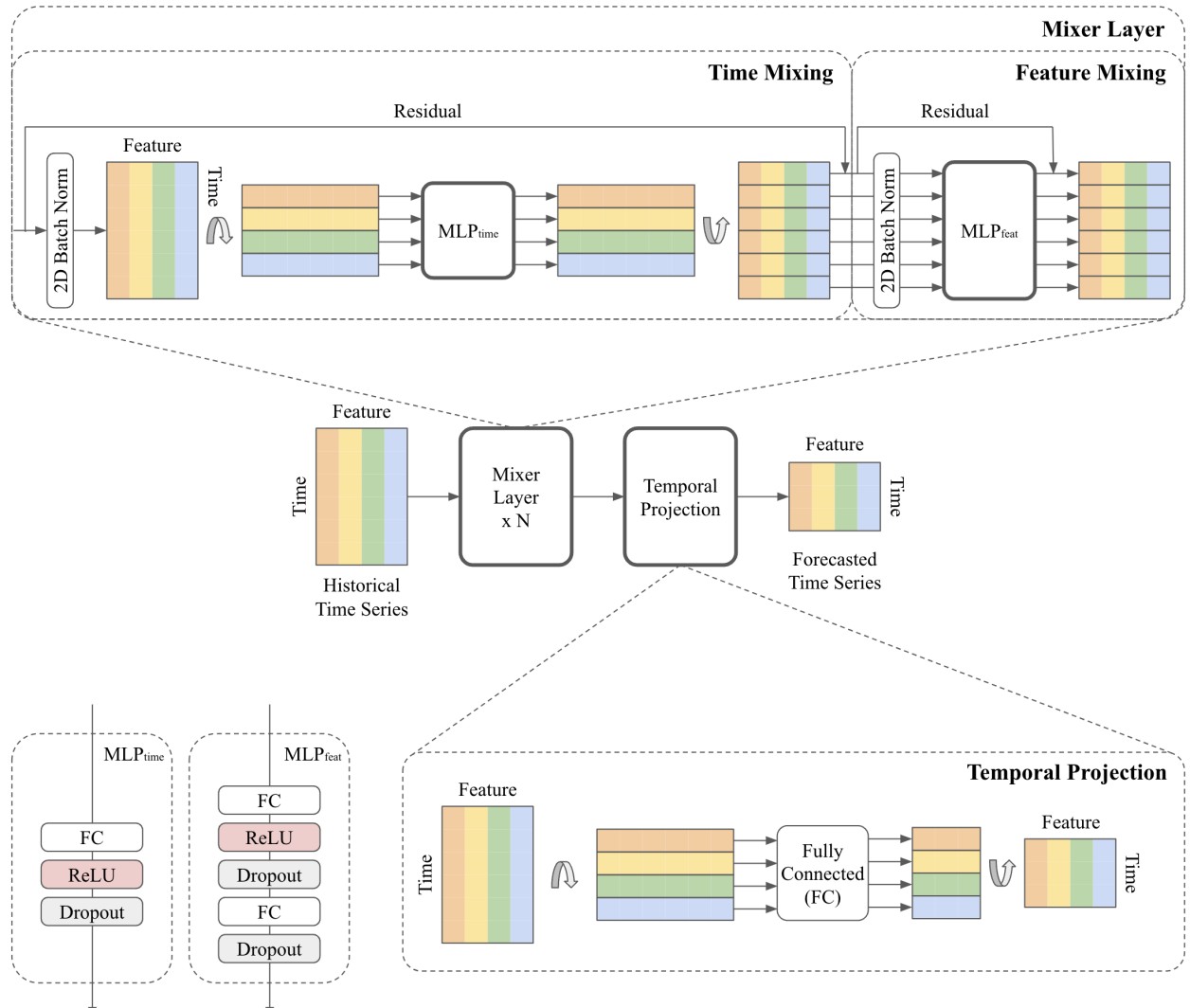

Figure 1: TSMixer for multivariate time series forecasting. The columns of the inputs means different features/variates and the rows are time steps. The fully-connected operations are row-wise. TSMixer contains interleaving time-mixing and feature-mixing MLPs to aggregate information. The number of mixer layer is denoted as $N$. The time-mixing MLPs are shared across all features and the feature-mixing MLPs are shared across all of the time steps. The design allow TSMixer to automatically adapt the use of both temporal and cross-variate information with limited number of parameters for superior generalization. The extension with auxiliary information is also explored in this paper.

- Persistent temporal patterns: encompassing trends and seasonal patterns, e.g., long-term inflation, day-of-week effects;

- Cross-variate information: correlations between different variables, e.g., an increase in blood pressure associated with a rise in body weight;

- Auxiliary features: comprising static features and future information, e.g., product categories and promotional events.

Traditional models, such as ARIMA (Box et al., 1970), are designed for univariate time series, where only temporal information is available. Therefore, they face limitations when dealing with challenging real-world data, which often contains complex cross-variate information and auxiliary features. In contrast, numerous deep learning models, particularly Transformer-based models, have been proposed due to their capacity to capture both complex temporal patterns and cross-variate dependencies (Gamboa, 2017; Li et al., 2019; Wen et al., 2017; Zhou et al., 2021; Wu et al., 2021; Lim & Zohren, 2021; Liu et al., 2022a; Zhou et al., 2022b; Liu et al., 2022b; Zhou et al., 2022a) .

The natural intuition is that multivariate models, such as those based on Transformer architectures, should be more effective than univariate models due to their ability to leverage cross-variate information. However, Zeng et al. (2023) revealed that this is not always the case – Transformer-based models can indeed be significantly worse than simple univariate temporal linear models on many commonly used forecasting benchmarks. The multivariate models seem to suffer from overfitting especially when the target time series is not correlated with other covariates. This surprising finding has raised two essential questions:

1. Does cross-variate information truly provide a benefit for time series forecasting?

2. When cross-variate information is not beneficial, can multivariate models still perform as well as univariate models?

To address these questions, we begin by analyzing the effectiveness of temporal linear models. Our findings indicate that their time-step-dependent characteristics render temporal linear models great candidates for learning temporal patterns under common assumptions. Consequently, we gradually increase the capacity of linear models by

1. stacking temporal linear models with non-linearities (TMix-Only),

2. introducing cross-variate feed-forward layers (TSMixer).

The resulting TSMixer alternatively applies MLPs across time and feature dimensions, conceptually corresponding to *time-mixing* and *feature-mixing* operations, efficiently capturing both temporal patterns and cross-variate information, as illustrated in Fig. 1. The residual designs ensure that TSMixer retains the capacity of temporal linear models while still being able to exploit cross-variate information.

We evaluate TSMixer on commonly used long-term forecasting datasets (Wu et al., 2021) where univariate models have outperformed multivariate models. Our ablation study demonstrates the effectiveness of stacking temporal linear models and validates that cross-variate information is less beneficial on these popular datasets, explaining the superior performance of univariate models. Even so, TSMixer is on par with state-of-the-art univariate models and significantly outperforms other multivariate models.

To demonstrate the benefit of multivariate models, we further evaluate TSMixer on the challenging M5 benchmark, a large-scale retail dataset used in the M-competition (Makridakis et al., 2022). M5 contains crucial cross-variate interactions such as sell prices (Makridakis et al., 2022). The results show that cross-variate information indeed brings significant improvement, and TSMixer can effectively leverage this information. Furthermore, we propose a principle design to extend TSMixer to handle auxiliary information such as static features and future time-varying features. It aligns the different types of features into the same shape then applied mixer layers on the concatenated features to leverage the interactions between them. In this more practical and challenging setting, TSMixer outperforms models that are popular in industrial applications, including DeepAR (Salinas et al. 2020, Amazon SageMaker) and TFT (Lim et al. 2021, Google Cloud Vertex), demonstrating its strong potential for real world impact.

We summarize our contributions as below:

- We analyze the effectiveness of state-of-the-art linear models and indicate that their time-step-dependent characteristics make them great candidates for learning temporal patterns under common assumptions.

Table 1: Recent works in time series forecasting. Category I is univariate time series forecasting; Category II is multivariate time series forecasting, and Category III is time series forecasting with auxiliary information. In this work, we propose TSMixer for Category II. We also extend TSMixer to leverage auxiliary information including static and future time-varying features for Category III.

| Category | Extrapolating temporal patterns | Consideration of cross-variate information (i.e. multivariateness) | Consideration of auxiliary features | Models |
|---|---|---|---|---|
| I | ✔ | | | ARIMA (Box et al., 1970) 
 N-BEATS (Oreshkin et al., 2020) 
 LTSF-Linear (Zeng et al., 2023) 
 PatchTST (Nie et al., 2023) |
| II | ✔ | ✔ | | Informer (Zhou et al., 2021) 
 Autoformer (Wu et al., 2021) 
 Pyraformer (Liu et al., 2022a) 
 FEDformer (Zhou et al., 2022b) 
 NS-Transformer (Liu et al., 2022b) 
 FiLM (Zhou et al., 2022a) 
 **TSMixer** (this work) |
| III | ✔ | ✔ | ✔ | MQRNN (Wen et al., 2017) 
 DSSM (Rangapuram et al., 2018) 
 DeepAR (Salinas et al., 2020) 
 TFT (Lim et al., 2021) 
 **TSMixer-Ext** (this work) |

- We propose TSMixer, an innovative architecture which retains the capacity of linear models to capture temporal patterns while still being able to exploit cross-variate information.

- We point out the potential risk of evaluating multivariate models on common long-term forecasting benchmarks.

- Our empirical studies demonstrate that TSMixer is the first multivariate model which is on par with univariate models on common benchmarks and achieves state-of-the-art on a large-scale industrial application where cross-variate information is crucial.

## 2 Related Work

Broadly, time series forecasting is the task of predicting future values of a variable or multiple related variables, given a set of historical observations. Deep neural networks have been widely investigated for this task (Zhang et al., 1998; Kourentzes, 2013; Lim & Zohren, 2021). In Table 1 we coarsely split notable works into three categories based on the information considered by the model: (I) univariate forecasting, (II) multivariate forecasting, and (III) multivariate forecasting with auxiliary information.

Multivariate time series forecasting with deep neural networks has been getting increasingly popular with the motivation that *modeling the complex relationships between covariates should improve the forecasting performance.* Transformer-based models (Category II) are common choices for this scenario because of their superior performance in modeling long and complex sequential data (Vaswani et al., 2017). Various variants of Transformers have been proposed to further improve efficiency and accuracy. Informer (Zhou et al., 2021) and Autoformer (Wu et al., 2021) tackle the efficiency bottleneck with different attention designs costing less memory usage for long-term forecasting. FEDformer (Zhou et al., 2022b) and FiLM (Zhou et al., 2022a) decompose the sequences using Fast Fourier Transformation for better extraction of long-term information. There are also extensions on improving specific challenges, such as non-stationarity (Kim et al., 2022; Liu et al., 2022b). Despite the advances in Transformer-based models for multivariate forecasting, Zeng et al. (2023) indeed show the counter-intuitive result that a simple univariate linear model (Category I), which

treats multivariate data as several univariate sequences, can outperform all of the proposed multivariate Transformer models by a significant margin on commonly-used long-term forecasting benchmarks. Similarly, Nie et al. (2023) advocate against modeling the cross-variate information and propose a univariate patch Transformer for multivariate forecasting tasks and show state-of-the-art accuracy on multiple datasets. As one of the core contributions, instead, we find that this conclusion mainly comes from the dataset bias, and might not generalize well to some real-world applications.

There are other works that consider a scenario when auxiliary information ((Category III)), such as static features (e.g. location) and future time-varying features (e.g. promotion in coming weeks), are available. Commonly used forecasting models have been extended to handle these auxiliary features. These include state-space models (Rangapuram et al., 2018; Alaa & van der Schaar, 2019; Gu et al., 2022), RNN variants Wen et al. (2017); Salinas et al. (2020), and attention models Lim et al. (2021). Most real-world time-series datasets are more aligned with this setting and that is why these deep learning models have achieved great success in various applications and are widely used in industry (e.g. DeepAR (Salinas et al., 2020) of AWS SageMaker and TFT (Lim et al., 2021) of Google Cloud Vertex). One drawback of these models is their complexity, particularly when compared to the aforementioned univariate models.

Our motivations for TSMixer stem from analyzing the performance of linear models for time series forecasting. Similar architectures have been considered for other data types before, for example the proposed TSMixer in a way resembles the well-known MLP Mixer architecture, from computer vision (Tolstikhin et al., 2021). Mixer models have also been applied to text (Fusco et al., 2022), speech (Tatanov et al., 2022), network traffic (Zheng et al., 2022) and point cloud (Choe et al., 2022). Yet, to the best of our knowledge, the use of an MLP Mixer based architecture for time series forecasting has not been explored in the literature.

## 3 Linear Models for Time Series Forecasting

The superiority of linear models over more complex sequential architectures, like Transformers, has been empirically demonstrated Zeng et al. (2023). We first provide theoretical insights on the capacity of linear models which might have been overlooked due to its simplicity compared to other sequential models. We then compare linear models with other architectures and show that linear models have a characteristic not present in RNNs and Transformers – they have the appropriate representation capacity to learn the time dependency for a univariate time series. This finding motivates the design of our proposed architecture, presented in Sec. 4.

**Notation:** Let the historical observations be $\boldsymbol{X} \in \mathbb{R}^{L \times C_x}$, where $L$ is the length of the lookback window and $C_x$ is the number of variables. We consider the task of predicting $\boldsymbol{Y} \in \mathbb{R}^{T \times C_y}$, where $T$ is the number of future time steps and $C_y$ is the number of time series we want to predict. In this work, we focus on the case when the past values of the target time series are included in the historical observation ($C_y \leq C_x$). A linear model learns parameters $\boldsymbol{A} \in \mathbb{R}^{T \times L}, \boldsymbol{b} \in \mathbb{R}^{T \times 1}$ to predict the values of the next $T$ steps as:

$$\hat{\boldsymbol{Y}} = \boldsymbol{A}\boldsymbol{X} \oplus \boldsymbol{b} \in \mathbb{R}^{T \times C_x}, \tag{1}$$

where $\oplus$ means column-wise addition. The corresponding $C_y$ columns in $\hat{\boldsymbol{Y}}$ can be used to predict $\boldsymbol{Y}$.

**Theoretical insights:** For time series forecasting, most impactful real-world applications have either smoothness or periodicity in them, as otherwise the predictability is low and the predictive models would not be reliable. First, we consider the common assumption that the time series is periodic (Holt, 2004; Zhang & Qi, 2005). Given an arbitrary periodic function $x(t) = x(t - P)$, where $P < L$ is the period. There is a solution of linear models to perfectly predict the future values as follows:

$$\boldsymbol{A}_{ij} = \begin{cases} 1, & \text{if } j = L - P + (i \bmod P) \\ 0, & \text{otherwise} \end{cases}, \boldsymbol{b}_i = 0. \tag{2}$$

When extending to affine-transformed periodic sequences, $x(t) = a \cdot x(t - P) + c$, where $a, c \in \mathbb{R}$ are constants, the linear model still has a solution for perfect prediction:

$$\boldsymbol{A}_{ij} = \begin{cases} a, & \text{if } j = L - P + (i \bmod P) \\ 0, & \text{otherwise} \end{cases}, \boldsymbol{b}_i = c. \tag{3}$$

A more general assumption is that the time series can be decomposed into a periodic sequence and a sequence with smooth trend (Holt, 2004; Zhang & Qi, 2005; Wu et al., 2021; Zhou et al., 2022b). In this case, we show the following property (see the proof in Appendix A):

**Theorem 3.1.** *Let $x(t) = g(t) + f(t)$, where $g(t)$ is a periodic signal with period $P$ and $f(t)$ is Lipschitz smooth with constant $K$ (i.e. $\left| \frac{f(a) - f(b)}{a - b} \right| \leq K$), then there exists a linear model with lookback window size $L \geq P + 1$ such that $|y_i - \hat{y}_i| \leq K(i + \min(i, P)), \forall i = 1, \ldots, T$.*

This derivation illustrates that linear models constitute strong candidates to capture temporal relationships. For the non-periodic patterns, as long as they are smooth, which is often the case in practice, the error is still bounded given an adequate lookback window size.

**Differences from conventional deep learning models.** Following the discussions in Zeng et al. (2023) and Nie et al. (2023), our analysis of linear models offers deeper insights into why previous deep learning models tend to overfit the data. Linear models possess a unique characteristic wherein the weights of the mapping are fixed for each time step in the input sequence. This "time-step-dependent" characteristic is a crucial component of our previous findings and stands in contrast to recurrent or attention-based architectures, where the weights over the input sequence are outputs of a "data-dependent" function, such as the gates in LSTMs or attention layers in Transformers. Time-step-dependent models vs. data-dependent models are illustrated in Fig. 2. The time-step-dependent linear model, despite its simplicity, proves to be highly effective in modeling temporal patterns. Conversely, even though recurrent or attention architectures have high representational capacity, achieving time-step independence is challenging for them. They usually overfit on the data instead of solely considering the positions. This unique property of linear models may help explain the results in Zeng et al. (2023), where no other method was shown to match the performance of the linear model.

**Limitations of the analysis.** The purpose of the analysis is to understand the effectiveness of temporal linear models in univariate scenario. Real-world time series data might have high volatility, making the patterns non-periodic and non-smooth. In such scenarios, relying solely on past-observed temporal patterns might be suboptimal. The analysis beyond Lipschitz cases could be challenging and out of the scope of this paper (Zhang, 2023), so we leave the analysis for more complex cases for the future work. Nevertheless, the analysis motivates us to develop a more powerful model based on linear models, which are introduced in Section 4. We also show the importance of effectively utilizing multivariate information as other covariates might contain the information that can be used to model volatility – indeed our results in Table 5 underline that.

## 4 TSMixer Architecture

Expanding upon our finding that linear models can serve as strong candidates for capturing time dependencies, we initially propose a natural enhancement by stacking linear models with non-linearities to form multi-layer perceptrons (MLPs). Common deep learning techniques, such as normalization and residual connections, are applied to facilitate efficient learning. However, this architecture does not take cross-variate information into account.

To better leverage cross-variate information, we propose the application of MLPs in the time-domain and the feature-domain in an alternating manner. The time-domain MLPs are shared across all of the features, while the feature-domain MLPs are shared across all of the time steps. This resulting model is akin to the MLP-Mixer architecture from computer vision (Tolstikhin et al., 2021), with time-domain and feature-domain

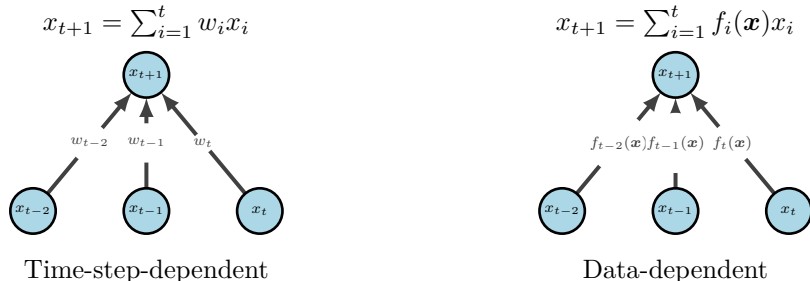

Figure 2: Illustrations of time-step-dependent and data-dependent models within a single forecasting time step.

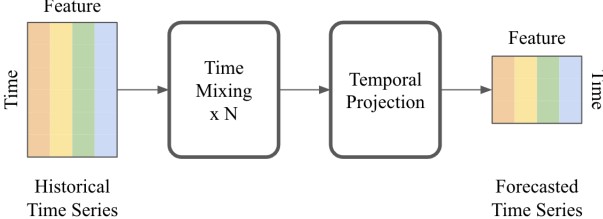

Figure 3: The architecture of TMix-Only. It is similar to TSMixer but only applies time-mixing.

operations representing time-mixing and feature-mixing operations, respectively. Consequently, we name our proposed architecture Time-Series Mixer (TSMixer).

The interleaving design between these two operations efficiently utilizes both temporal dependencies and cross-variate information while limiting computational complexity and model size. It allows TSMixer to use a long lookback window (see Sec. 3), while maintaining the parameter growth in only $O(L + C)$ instead of $O(LC)$ if fully-connected MLPs were used. To better understand the utility of cross-variate information and feature-mixing, we also consider a simplified variant of TSMixer that only employs time-mixing, referred to as TMix-Only, which consists of a residual MLP shared across each variate, as illustrated in Fig. 3. We also present the extension of TSMixer to scenarios where auxiliary information about the time series is available.

## 4.1 TSMixer for Multivariate Time Series Forecasting

For multivariate time series forecasting where only historical data are available, TSMixer applies MLPs alternatively in time and feature domains. The architecture is illustrated in Fig. 1. TSMixer comprises the following components:

- **Time-mixing MLP**: Time-mixing MLPs model temporal patterns in time series. They consist of a fully-connected layer followed by an activation function and dropout. They transpose the input to apply the fully-connected layers along the time domain and shared by features. We employ a single-layer MLP, as demonstrated in Sec.3, where a simple linear model already proves to be a strong model for learning complex temporal patterns.

- **Feature-mixing MLP**: Feature-mixing MLPs are shared by time steps and serve to leverage covariate information. Similar to Transformer-based models, we consider two-layer MLPs to learn complex feature transformations.

- **Temporal Projection**: Temporal projection, identical to the linear models inZeng et al. (2023), is a fully-connected layer applied on time domain. They not only learn the temporal patterns but also map the time series from the original input length $L$ to the target forecast length $T$.

- **Residual Connections**: We apply residual connections between each time-mixing and feature-mixing layer. These connections allow the model to learn deeper architectures more efficiently and allow the model to effectively ignore unnecessary time-mixing and feature-mixing operations.

- **Normalization**: Normalization is a common technique to improve deep learning model training. While the preference between batch normalization and layer normalization is task-dependent, Nie et al. (2023) demonstrates the advantages of batch normalization on common time series datasets. In contrast to typical normalization applied along the feature dimension, we apply 2D normalization on both time and feature dimensions due to the presence of time-mixing and feature-mixing operations.

Contrary to some recent Transformer advances with increased complexity, the architecture of TSMixer is relatively simple to implement. Despite its simplicity, we demonstrate in Sec. 5 that TSMixer remains competitive with state-of-the-art models at representative benchmarks.

### 4.2 Extended TSMixer for Time Series Forecasting with Auxiliary Information

In addition to the historical observations, many real-world scenarios allow us to have access to static $\boldsymbol{S} \in \mathbb{R}^{1 \times C_s}$ (e.g. location) and future time-varying features $\boldsymbol{Z} \in \mathbb{R}^{T \times C_z}$ (e.g. promotion in subsequent weeks). The problem can also be extended to multiple time series, represented by $\boldsymbol{X}^{(i)}{}_{i=1}^{M}$, where $M$ is the number of time series, with each time series is associated with its own set of features. Most recent work, especially those focus on long-term forecasting, only consider the historical features and targets on all variables (i.e. $C_x = C_y > 1, C_s = C_z = 0$). In this paper, we also consider the case where auxiliary information is available (i.e. $C_s > 0, C_z > 0$).

To leverage the different types of features, we propose a principle design that naturally leverages the feature mixing to capture the interaction between them. We first design the align stage to project the feature with different shapes into the same shape. Then we can concatenate the features and seamlessly apply feature mixing on them. We extend TSMixer as illustrated in Fig. 4. The architecture comprises two parts: align and mixing. In the align stage, TSMixer aligns historical features ($\mathbb{R}^{L \times C_x}$) and future features ($\mathbb{R}^{T \times C_z}$) into the same shape ($\mathbb{R}^{L \times C_h}$) by applying temporal projection and a feature-mixing layer, where $C_h$ represents the size of hidden layers. Additionally, it repeats the static features to transform their shape from $\mathbb{R}^{1 \times C_s}$ to $\mathbb{R}^{T \times C_s}$ in order to align the output length.

In the mixing stage, the mixing layer, which includes time-mixing and feature-mixing operations, naturally leverages temporal patterns and cross-variate information from all features collectively. Lastly, we employ a fully-connected layer to generate outputs for each time step. The outputs can either be real values of the forecasted time series ($\mathbb{R}^{T \times C_y}$), typically optimized by mean absolute error or mean square error, or in some tasks, they may generate parameters of a target distribution, such as negative binomial distribution for retail demand forecasting (Salinas et al., 2020). We slightly modify mixing layers to better handle M5 dataset, as described in Appendix B.

### 4.3 Differences between TSMixer and MLP-Mixer

While TSMixer shares architectural similarities with MLP-Mixer, the development of TSMixer, motivated by our analysis in Section 3, has led to a unique normalization approach. In TSMixer, two dimensions represent features and time steps, unlike MLP-Mixer's features and patches. Consequently, we apply 2D normalization to maintain scale across features and time steps, since we have discovered the importance of utilizing temporal patterns in forecasting. Besides, we have proposed an extended version of TSMixer to better extract information from heterogeneous inputs, essential to achieve state-of-the-art results in real-world scenarios.

## 5 Experiments

We evaluate TSMixer on seven popular multivariate long-term forecasting benchmarks and a large-scale real-world retail dataset, M5 (Makridakis et al., 2022). The long-term forecasting datasets cover various

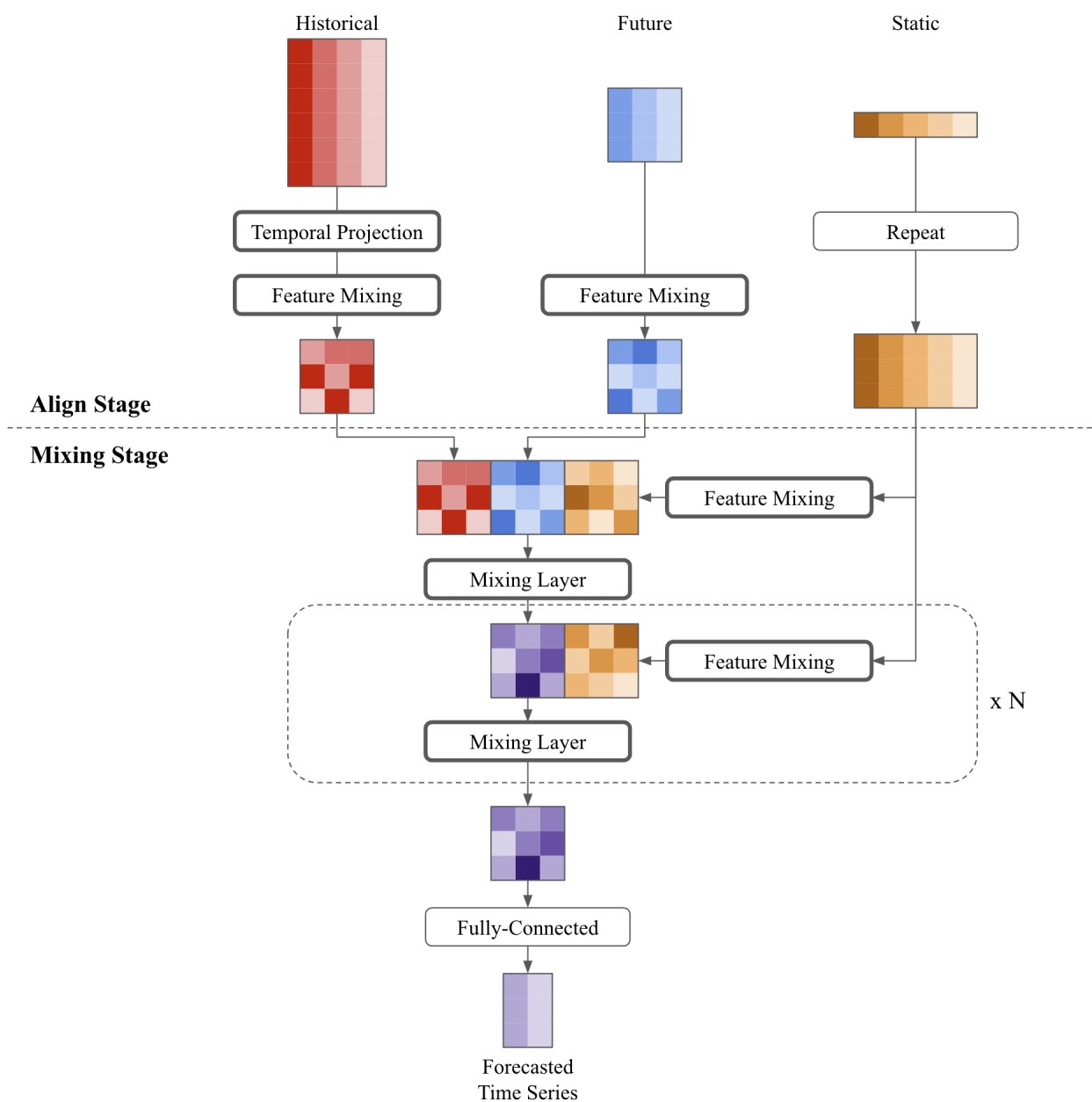

Figure 4: TSMixer with auxiliary information. The columns of the inputs are features and the rows are time steps. We first align the sequence lengths of different types of inputs to concatenate them. Then we apply mixing layers to model their temporal patterns and cross-variate information jointly.

applications such as weather, electricity, and traffic, and are comprised of multivariate time series without auxiliary information. The M5 dataset is for the competition task of predicting the sales of various items at Walmart. It is a large scale dataset containing 30,490 time series with static features such as store locations, as well as time-varying features such as campaign information. This complexity renders M5 a more challenging benchmark to explore the potential benefits of cross-variate information and auxiliary features. The statistics of these datasets are presented in Table 2.

Table 2: Statistics of all datasets. Note that Electricity and Traffic can be considered as multivariate time series or multiple univariate time series since all variates share the same physical meaning in the dataset (e.g. electricity consumption at different locations).

| | ETTh1/h2 | ETTm1/m2 | Weather | Electricity | Traffic | M5 |
|---|---|---|---|---|---|---|
| # of time series ($M$) | 1 | 1 | 1 | 1 | 1 | 30,490 |
| # of variants ($C$) | 7 | 7 | 21 | 321 | 862 | 1 |
| Time steps | 17,420 | 699,680 | 52,696 | 26,304 | 17,544 | 1,942 |
| Granularity | 1 hour | 15 minutes | 10 minutes | 1 hour | 1 hour | 1day |
| Historical feature ($C_x$) | 0 | 0 | 0 | 0 | 0 | 14 |
| Future feature ($C_z$) | 0 | 0 | 0 | 0 | 0 | 13 |
| Static feature ($C_s$) | 0 | 0 | 0 | 0 | 0 | 6 |
| Data partition (Train/Validation/Test) | 12/4/4 (month) | | 7:2:1 | | | 1886/28/28 (day) |

Table 3: Evaluation results on the long-term forecasting datasets. The numbers of models marked with "*" are obtained from Nie et al. (2023). The best numbers in each row are shown in **bold** and the second best numbers are underlined. We skip TMix-Only in comparisons as it performs similar to TSMixer. The last row shows the average percentage of MSE improvement of TSMixer over other methods.

| | | Multivariate Model | | | | | | | | | | Univariate Model | | | | | |
|---|---|---|---|---|---|---|---|---|---|---|---|---|---|---|---|---|---|
| Models | | **TSMixer** | | TFT | | FEDformer* | | Autoformer* | | Informer* | | **TMix-Only** | | Linear | | PatchTST* | |
| Metric | | MSE | MAE | MSE | MAE | MSE | MAE | MSE | MAE | MSE | MAE | MSE | MAE | MSE | MAE | MSE | MAE |
| ETTh1 | 96 | **0.361** | **0.392** | 0.674 | 0.634 | 0.376 | 0.415 | 0.435 | 0.446 | 0.941 | 0.769 | 0.359 | 0.391 | 0.368 | **0.392** | 0.370 | 0.400 |
| | 192 | **0.404** | 0.418 | 0.858 | 0.704 | 0.423 | 0.446 | 0.456 | 0.457 | 1.007 | 0.786 | 0.402 | 0.415 | **0.404** | **0.415** | 0.413 | 0.429 |
| | 336 | **0.420** | **0.431** | 0.900 | 0.731 | 0.444 | 0.462 | 0.486 | 0.487 | 1.038 | 0.784 | 0.420 | 0.434 | 0.436 | 0.439 | 0.422 | 0.440 |
| | 720 | 0.463 | 0.472 | 0.745 | 0.666 | 0.469 | 0.492 | 0.515 | 0.517 | 1.144 | 0.857 | 0.453 | 0.467 | 0.481 | 0.495 | **0.447** | **0.468** |
| ETTh2 | 96 | **0.274** | 0.341 | 0.409 | 0.505 | 0.332 | 0.374 | 0.332 | 0.368 | 1.549 | 0.952 | 0.275 | 0.342 | 0.297 | 0.363 | **0.274** | **0.337** |
| | 192 | **0.339** | 0.385 | 0.953 | 0.651 | 0.407 | 0.446 | 0.426 | 0.434 | 3.792 | 1.542 | 0.339 | 0.386 | 0.398 | 0.429 | 0.341 | **0.382** |
| | 336 | 0.361 | 0.406 | 1.006 | 0.709 | 0.400 | 0.447 | 0.477 | 0.479 | 4.215 | 1.642 | 0.366 | 0.413 | 0.500 | 0.491 | **0.329** | **0.384** |
| | 720 | 0.445 | 0.470 | 1.187 | 0.816 | 0.412 | 0.469 | 0.453 | 0.490 | 3.656 | 1.619 | 0.437 | 0.465 | 0.795 | 0.633 | **0.379** | **0.422** |
| ETTm1 | 96 | **0.285** | **0.339** | 0.752 | 0.626 | 0.326 | 0.390 | 0.510 | 0.492 | 0.626 | 0.560 | 0.284 | 0.338 | 0.303 | 0.346 | 0.293 | 0.346 |
| | 192 | **0.327** | **0.365** | 0.752 | 0.649 | 0.365 | 0.415 | 0.514 | 0.495 | 0.725 | 0.619 | 0.324 | 0.362 | 0.335 | **0.365** | 0.333 | 0.370 |
| | 336 | **0.356** | **0.382** | 0.810 | 0.674 | 0.392 | 0.425 | 0.510 | 0.492 | 1.005 | 0.741 | 0.359 | 0.384 | 0.365 | 0.384 | 0.369 | 0.392 |
| | 720 | 0.419 | **0.414** | 0.849 | 0.695 | 0.446 | 0.458 | 0.527 | 0.493 | 1.133 | 0.845 | 0.419 | 0.414 | 0.419 | 0.415 | **0.416** | 0.420 |
| ETTm2 | 96 | **0.163** | **0.252** | 0.386 | 0.472 | 0.180 | 0.271 | 0.205 | 0.293 | 0.355 | 0.462 | 0.162 | 0.249 | 0.170 | 0.266 | 0.166 | 0.256 |
| | 192 | **0.216** | **0.290** | 0.739 | 0.626 | 0.252 | 0.318 | 0.278 | 0.336 | 0.595 | 0.586 | 0.220 | 0.293 | 0.236 | 0.317 | 0.223 | 0.296 |
| | 336 | **0.268** | **0.324** | 0.477 | 0.494 | 0.324 | 0.364 | 0.343 | 0.379 | 1.270 | 0.871 | 0.269 | 0.326 | 0.308 | 0.369 | 0.274 | 0.329 |
| | 720 | 0.420 | 0.422 | 0.523 | 0.537 | 0.410 | 0.420 | 0.414 | 0.419 | 3.001 | 1.267 | 0.358 | 0.382 | 0.435 | 0.449 | **0.362** | **0.385** |
| Weather | 96 | **0.145** | **0.198** | 0.441 | 0.474 | 0.238 | 0.314 | 0.249 | 0.329 | 0.354 | 0.405 | 0.145 | 0.196 | 0.170 | 0.229 | 0.149 | **0.198** |
| | 192 | **0.191** | 0.242 | 0.699 | 0.599 | 0.275 | 0.329 | 0.325 | 0.370 | 0.419 | 0.434 | 0.190 | 0.240 | 0.213 | 0.268 | 0.194 | **0.241** |
| | 336 | **0.242** | **0.280** | 0.693 | 0.596 | 0.339 | 0.377 | 0.351 | 0.391 | 0.583 | 0.543 | 0.240 | 0.279 | 0.257 | 0.305 | 0.245 | 0.282 |
| | 720 | 0.320 | 0.336 | 1.038 | 0.753 | 0.389 | 0.409 | 0.415 | 0.426 | 0.916 | 0.705 | 0.325 | 0.339 | 0.318 | 0.356 | **0.314** | **0.334** |
| Electricity | 96 | 0.131 | 0.229 | 0.295 | 0.376 | 0.186 | 0.302 | 0.196 | 0.313 | 0.304 | 0.393 | 0.132 | 0.225 | 0.135 | 0.232 | **0.129** | **0.222** |
| | 192 | 0.151 | 0.246 | 0.327 | 0.397 | 0.197 | 0.311 | 0.211 | 0.324 | 0.327 | 0.417 | 0.152 | 0.243 | 0.149 | 0.246 | **0.147** | **0.240** |
| | 336 | **0.161** | 0.261 | 0.298 | 0.380 | 0.213 | 0.328 | 0.214 | 0.327 | 0.333 | 0.422 | 0.166 | 0.260 | 0.164 | 0.263 | 0.163 | **0.259** |
| | 720 | **0.197** | 0.293 | 0.338 | 0.412 | 0.233 | 0.344 | 0.236 | 0.342 | 0.351 | 0.427 | 0.200 | 0.291 | 0.199 | 0.297 | **0.197** | **0.290** |
| Traffic | 96 | 0.376 | 0.264 | 0.678 | 0.362 | 0.576 | 0.359 | 0.597 | 0.371 | 0.733 | 0.410 | 0.370 | 0.258 | 0.395 | 0.274 | **0.360** | **0.249** |
| | 192 | 0.397 | 0.277 | 0.664 | 0.355 | 0.610 | 0.380 | 0.607 | 0.382 | 0.777 | 0.435 | 0.390 | 0.268 | 0.406 | 0.279 | **0.379** | **0.256** |
| | 336 | 0.413 | 0.290 | 0.679 | 0.354 | 0.608 | 0.375 | 0.623 | 0.387 | 0.776 | 0.434 | 0.404 | 0.276 | 0.416 | 0.286 | **0.392** | **0.264** |
| | 720 | 0.444 | 0.306 | 0.610 | 0.326 | 0.621 | 0.375 | 0.639 | 0.395 | 0.827 | 0.466 | 0.443 | 0.297 | 0.454 | 0.308 | **0.432** | **0.286** |
| **TSMixer MSE Imp.** | | | | **51.94%** | | **16.69%** | | **24.51%** | | **62.40%** | | -0.66% | | **6.77%** | | -1.53% | |

For multivariate long-term forecasting datasets, we follow the settings in recent research (Liu et al., 2022b; Zhou et al., 2022a; Nie et al., 2023). We set the input length $L = 512$ as suggested in Nie et al. (2023) and evaluate the results for prediction lengths of $T = \{96, 192, 336, 720\}$. We use the Adam optimization algorithm (Kingma & Ba, 2015) to minimize the mean square error (MSE) training objective, and consider MSE

and mean absolute error (MAE) as the evaluation metrics. We apply reversible instance normalization (Kim et al., 2022) to ensure a fair comparison with the state-of-the-art PatchTST (Nie et al., 2023).

For the M5 dataset, we mostly follow the data processing from Alexandrov et al. (2020). We consider the prediction length of $T = 28$ (same as the competition), and set the input length to $L = 35$. We optimize log-likelihood of negative binomial distribution as suggested by Salinas et al. (2020). We follow the competition's protocol (Makridakis et al., 2022) to aggregate the predictions at different levels and evaluate them using the weighted root mean squared scaled error (WRMSSE). More details about the experimental setup and hyperparameter tuning can be found in Appendices C and E.

## 5.1 Multivariate Long-term Forecasting

For multivariate long-term forecasting tasks, we compare TSMixer to state-of-the-art multivariate models such as FEDformer (Zhou et al., 2022b), Autoformer (Wu et al., 2021), Informer (Zhou et al., 2021), and univariate models like PatchTST (Nie et al., 2023) and LTSF-Linear (Zeng et al., 2023). Additionally, we include TFT (Lim et al., 2021), a deep learning-based model that considers auxiliary information, as a baseline to understand the limitations of solely relying on historical features. We also evaluate TMix-Only, a variant of TSMixer that only applies time-mixing, to assess the effectiveness of feature-mixing. The results are presented in Table 3. A comparison with other MLP-like alternatives is provided in Appendix F.

**TMix-Only**   We first examine the results of univariate models. Compared to the linear model, TMix-Only shows that stacking proves beneficial, even without considering cross-variate information. Moreover, TMix-Only performs at a level comparable to the state-of-the-art PatchTST, suggesting that the simple time-mixing layer is on par with more complex attention mechanisms.

**TSMixer**   Our results indicate that TSMixer exhibits similar performance to TMix-Only and PatchTST. It significantly outperforms state-of-the-art multivariate models and achieves competitive performance compared to PatchTST, the state-of-the-art univariate model. TSMixer is the only multivariate model that is competitive with univariate models with all other multivariate models performing significantly worse than univariate models. The performance of TSMixer is also similar to that of TMix-Only, which implies that feature-mixing is not beneficial for these benchmarks. These observations are consistent with findings in (Zeng et al., 2023) and (Nie et al., 2023). The results suggest that cross-variate information may be less significant in these datasets, indicating that the commonly used datasets may not be sufficient to evaluate a model's capability of utilizing covariates. However, we will demonstrate that cross-variate information can be useful in other scenarios.

**Effects of lookback window length**   To gain a deeper understanding of TSMixer's capacity to leverage longer sequences, we conduct experiments with varying lookback window sizes, specifically $L = \{96, 336, 512, 720\}$. We also perform similar experiments on linear models to support our findings presented in Section 3. The results of these experiments are depicted in Fig. 5. More results and details can be found in Appendix D. Our empirical analyses reveal that the performance of linear models improves significantly as the lookback window size increases from 96 to 336, and appears to be reaching a convergence point at 720. This aligns with our prior findings that the performance of linear models is dependent on the lookback window size. On the other hand, TSMixer achieves the best performance when the window size is set to 336 or 512, and maintains the similar level of performance as the window size is increased to 720. As noted by Nie et al. (2023), many multivariate Transformer-based models (such as Transformer, Informer, Autoformer, and FEDformer) do not benefit from lookback window sizes greater than 192, and are prone to overfitting when the window size is increased. In comparison, TSMixer demonstrates a superior ability to leverage longer sequences and better generalization capabilities than other multivariate models.

## 5.2 Large-scale Demand Forecasting

We evaluate TSMixer on the large-scale retail dataset M5 to explore the model's ability to leverage complicated cross-variate information and auxiliary features. M5 comprises thousands of multivariate time series, each with its own historical observations, future time-varying features, and static features, in contrast to the long-term

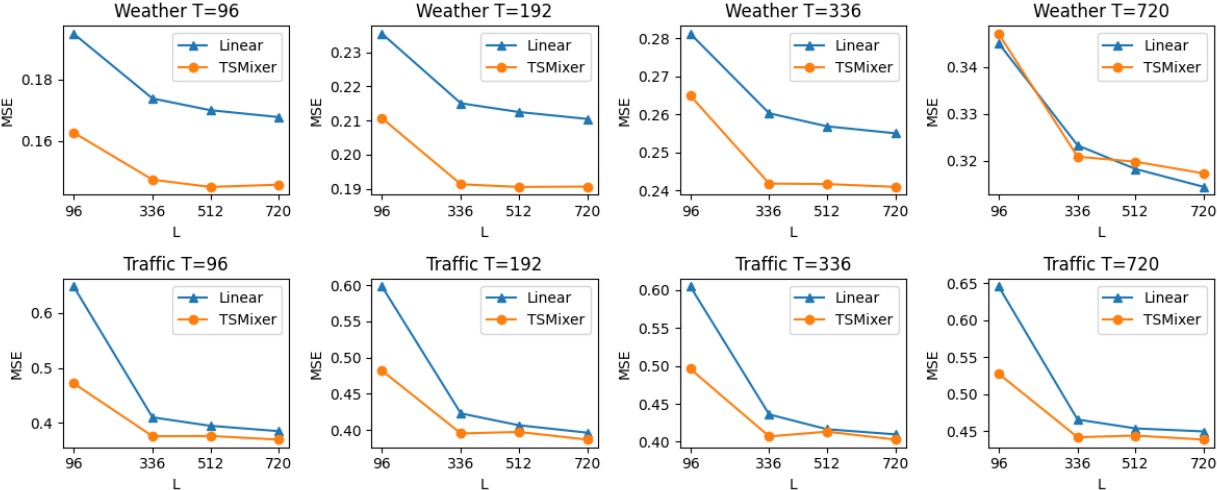

Figure 5: Performance comparison on varying lookback window size $L$ of linear models and TSMixer.

Table 4: Evaluation on M5 without auxiliary information. We report the mean and standard deviation of WRMSSE across 5 different random seeds. TMix-Only is a univariate variant of TSMixer where only time-mixing is applied. The multivariate models outperforms univariate models with a significant gap.

| Models | Multivariate | Test WRMSSE | Val WRMSSE |
|---|---|---|---|
| Linear | | 0.983±0.016 | 1.045±0.018 |
| PatchTST | | 0.976±0.014 | 0.992±0.011 |
| **TMix-Only** | | 0.960±0.041 | 1.000±0.027 |
| Autoformer | ✔ | 0.742±0.029 | 0.640±0.023 |
| FEDformer | ✔ | 0.804±0.039 | 0.674±0.014 |
| **TSMixer** | ✔ | **0.737±0.033** | 0.605±0.027 |

forecasting benchmarks, which typically consist of a single multivariate historical time series. We utilize TSMixer-Ext, the architecture introduced in Sec.4.2, to leverage the auxiliary information. Furthermore, the presence of a high proportion of zeros in the target sequence presents an additional challenge for prediction. Therefore, we learn negative binomial distributions, as suggested bySalinas et al. (2020), to better fit the distribution.

**Forecast with Historical Features Only** First, we compare TSMixer with other baselines using historical features only. As shown in Table 4 the multivariate models perform much better than univariate models for this dataset. Notably, PatchTST, which is designed to ignore cross-variate information, performs significantly

Table 5: Evaluation on M5 with auxiliary information.

| Models | Auxiliary feature | | Test WRMSSE | Val WRMSSE |
|---|---|---|---|---|
| | Static | Future | | |
| DeepAR | ✔ | ✔ | 0.789±0.025 | 0.611±0.007 |
| TFT | ✔ | ✔ | 0.670±0.020 | 0.579±0.011 |
| TSMixer-Ext | | | 0.737±0.033 | 0.000±0.000 |
| | ✔ | | 0.657±0.046 | 0.000±0.000 |
| | | ✔ | 0.697±0.028 | 0.000±0.000 |
| | ✔ | ✔ | **0.640**±0.013 | 0.568±0.009 |

Table 6: Computational cost on M5. All models are trained on a single NVIDIA Tesla V100 GPU. All models are implemented in PyTorch, except TFT, which is implemented in MXNet.

| Models | Multivariate | Auxiliary feature | # of params | training time (s) | inference (step/s) |
|---|:---:|:---:|---|---|---|
| Linear | | | 1K | 2958.18 | 110 |
| PatchTST | | | 26.7K | 886.101 | 120 |
| **TMix-Only** | | | 6.3K | 4073.72 | 110 |
| Autoformer | ✔ | | 471K | 119087.64 | 42 |
| FEDformer | ✔ | | 1.7M | 11084.43 | 56 |
| **TSMixer** | ✔ | | 189K | 11077.95 | 96 |
| DeepAR | ✔ | ✔ | 1M | 8743.55 | 105 |
| TFT | ✔ | ✔ | 2.9M | 14426.79 | 22 |
| **TSMixer-Ext** | ✔ | ✔ | 244K | 11615.87 | 108 |

worse than multivariate TSMixer and FEDformer. This result underscores the importance of modeling cross-variate information on some forecasting tasks, as opposed to the argument in (Nie et al., 2023). Furthermore, TSMixer substantially outperforms FEDformer, a state-of-the-art multivariate model.

TSMixer exhibits a unique value as it is the only model that performs as well as univariate models when cross-variate information is not useful, and it is the best model to leverage cross-variate information when it is useful.

**Forecast with Auxiliary Information**   To understand the extent to which TSMixer can leverage auxiliary information, we compare TSMixer against established time series forecasting algorithms, TFT (Lim et al., 2021) and DeepAR (Salinas et al., 2020). Table 5 shows that with auxiliary features TSMixer outperforms all other baselines by a significant margin. This result demonstrates the superior capability of TSMixer for modeling complex cross-variate information and effectively leveraging auxiliary features, an impactful capability for real-world time-series data beyond long-term forecasting benchmarks. We also conduct ablation studies by removing the static features and future time-varying features. The results demonstrates that while the impact of static features is more prominent, both static and future time-varying features contribute to the overall performance of TSMixer. This further emphasizes the importance of incorporating auxiliary features in time series forecasting models.

**Computational Cost**   We measure the computational cost of each models with their best hyperparameters on M5. As shown in Table 6, TSMixer has much smaller size compared to RNN- and Transformer-based models. TSMixer has similar training time with multivariate models, however, it achieves much faster inference, which is almost the same as simple linear models. Note that PatchTST has faster inference speed because it merges the feature dimension into the batch dimension, which leads to more parallelism but loses the multivariate information, a key aspect for high forecasting accuracy on real-world time-series data.

## 6   Conclusions

We propose TSMixer, a novel architecture for time series forecasting that is designed using MLPs instead of commonly used RNNs and attention mechanisms to obtain superior generalization with a simple architecture. Our results at a wide range of real-world time series forecasting tasks demonstrate that TSMixer is highly effective in both long-term forecasting benchmarks for multivariate time-series, and real-world large-scale retail demand forecasting tasks. Notably, TSMixer is the only multivariate model that is able to achieve similar performance to univariate models in long term time series forecasting benchmarks. The TSMixer architecture has significant potential for further improvement and we believe it will be useful in a wide range of time series forecasting tasks. Some of the potential future works include further exploring the interpretability of TSMixer, as well as its scalability to even larger datasets. We hope this work will pave the way for more innovative architectures for time series forecasting.

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

## A Proof of Theorem 3.1

**Theorem 3.1.** *Let $x(t) = g(t) + f(t)$, where $g(t)$ is a periodic signal with period $P$ and $f(t)$ is Lipschitz smooth with constant $K$ (i.e. $\left|\frac{f(a)-f(b)}{a-b}\right| \leq K$), then there exists a linear model with lookback window size $L \geq P + 1$ such that $|y_i - \hat{y}_i| \leq K(i + \min(i, P)), \forall i = 1, \ldots, T$.*

*Proof.* Without loss of generality, we assume the lookback window starts at $t = 1$ and the historical values is $\boldsymbol{x} \in \mathbb{R}^L$. The ground truth of the future time series:

$$y_i = x(L + i) = x(P + 1 + i) = g(P + 1 + i) + f(P + 1 + i) = g(1 + i) + f(P + 1 + i)$$

Let $\boldsymbol{A} \in \mathbb{R}^{T \times (P+1)}$, and

$$\boldsymbol{A}_{ij} = \begin{cases} 1, & \text{if } j = P + 1 \text{ or } j = (i \bmod P) + 1 \\ -1, & \text{if } j = 1 \\ 0, & \text{otherwise} \end{cases} , \boldsymbol{b}_i = 0$$

Then

$$\begin{aligned} \hat{y}_i &= \boldsymbol{A}_i \boldsymbol{x} + \boldsymbol{b} \\ &= x_{(i \bmod P)+1} - x_1 + x_{P+1} \\ &= x((i \bmod P) + 1) - x(1) + x(P + 1) \end{aligned}$$

So we have:

$$\begin{aligned} y_i - \hat{y}_i &= x(P + i + 1) - x((i \bmod P) + 1) + x(1) - x(P + 1) \\ &= (x(P + i + 1) - x((i \bmod P) + 1)) + (x(1) - x(P + 1)) \\ &= f(P + i + 1) - f((i \bmod P) + 1) + g(P + i + 1) - g((i \bmod P) + 1) \\ &\quad + f(1) - f(P + 1) + g(1) - g(P + 1) \\ &= (f(P + i + 1) - f(P + 1)) - (f((i \bmod P) + 1) - f(1)) \end{aligned}$$

And the mean absolute error between $y_i$ and $\hat{y}_i$ would be:

$$\begin{aligned} |y_i - \hat{y}_i| &= |(f(P + i + 1) - f(P + 1)) - (f((i \bmod P) + 1) - f(1))| \\ &\leq |f(P + i + 1) - f(P + 1)| + |f((i \bmod P) + 1) - f(1)| \\ &\leq K|(P + i + 1) - (P + 1)| + K|(i \bmod P + 1) - 1| \\ &\leq K(i + \min(i, P)) \end{aligned}$$

$\square$

## B Implementation Details

### B.1 Normalization

There are three types of normalizations used in the implementation:

1. Global normalization: Global normalization standardizes all variates of time series independently as a data pre-processing. The standardized data is then used for training and evaluation. It is a common setup in long-term time series forecasting experiments to prevent from the affects of different variate scales. For M5, since there is only one target time series (sales), we do not apply the global normalization.

2. Local normalization: In contrast to global normalization, local normalization is applied on each batch as pre-processing or post-processing. For long-term forecasting datasets, we apply reversible instance normalization (Kim et al., 2022) to ensure a fair comparison with the state-of-the-art results. In M5, we independently scale the sales of all products by their mean for model input and re-scale the model output.

3. Model-level normalization: We apply batch normalization on long-term forecasting datasets as suggested in (Nie et al., 2023) and apply layer normalization on M5 as described below.

## B.2 Differences between TSMixer and TSMixer-Ext

Due to the different normalizations between long-term forecasting benchmarks and M5, we slightly modify the mixing layers in TSMixer-Ext to better fit M5. We consider post-normalization rather than pre-normalization (Xiong et al., 2020) because pre-normalization may lead to NaN when the scale of input is too large. Furthermore, we apply layer normalization instead of batch normalization because batch normalization requires much larger batch size to obtain stable statistics of M5. The resulting architecture is shown in Fig. 6.

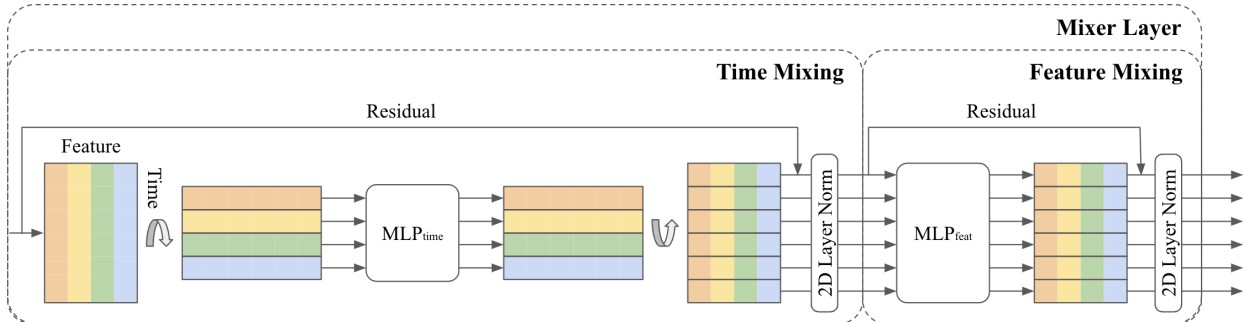

Figure 6: Mixing layers in TSMixer-Ext.

## B.3 Formulae of TSMixer architecture

In this section, we provide the mathematical formulae of each components in TSMixer.

### B.3.1 TSMixer Components

The TSMixer architecture is composed of several key components, which are implemented using a combination of linear layers, nonlinear activation functions, dropout, normalization, and residual connections. These are all standard deep learning operations that are commonly used. The major components of TSMixer are:

1. **Temporal Projection** and **Time Mixing**, which are used to model transformations between time steps.

2. **Feature Mixing**, which is used to model feature transformations.

3. **Conditional Feature Mixing**, which is used to transform hidden features based on the static features $S$.

4. **Mixer Layer**, which is the composition of the Time Mixing and the Feature Mixing.

5. **Conditional Mixer Layer**, which is the composition of the Time Mixing and the Conditional Feature Mixing.

For the layers involving the change of output size, we use subscripts $A \rightarrow B$ denotes the size is changing from $A$ to $B$.

**Temporal Projection** Given an input matrix $\boldsymbol{X} \in \mathbb{R}^{L \times C}$, the Temporal Projection (TP) is a linear layer that acts on the columns of $\boldsymbol{X}$ (denoted as $\boldsymbol{X}_{*,i}$) and is shared across all columns to project the time series from the input length to the prediction length. The operation is defined as:

$$\text{TP}_{L \to T}(\boldsymbol{X})_{*,i} = \boldsymbol{W}_1 \boldsymbol{X}_{*,i} + \boldsymbol{b}_1, \forall i = 1, \dots, C, \tag{4}$$

where $\boldsymbol{W}_1 \in \mathbb{R}^{L \times T}$ and $\boldsymbol{b}_1 \in \mathbb{R}^T$ are the weights and biases of the linear layer respectively. The subscript $L \to T$ denotes the mapping between input and output dimensions.

**Time Mixing** Similar to the Temporal Projection, the Time Mixing (TM) acts on all columns of $\boldsymbol{X}$ and applies commonly used deep learning layers to perform temporal feature transformation. The operation is defined as:

$$\text{TM}(\boldsymbol{X})_{*,i} =$$
$$\text{Norm}\left(\boldsymbol{X}_{*,i} + \text{Drop}\left(\sigma\left(\text{TP}_{L \to L}(\boldsymbol{X})_{*,i}\right)\right)\right),$$
$$\forall i = 1, \dots, C, \tag{5}$$

where $\sigma(\cdot)$ is an activation function, $\text{Drop}(\cdot)$ is dropout and $\text{Norm}(\cdot)$ can be layer normalization or batch normalization. It is important to note that the normalization is applied on the entire matrix (along both time and feature domain), rather than row-by-row (along the feature domain) as in Transformer-based models. The TM block allows TSMixer to effectively capture temporal dependencies in the time series data.

**Feature Mixing** The Feature Mixing (FM) is a two-layer residual MLP that acts on the rows of the input matrix $\boldsymbol{X} \in \mathbb{R}^{L \times C}$ and is shared across all rows. The block is designed to model feature transformations and is applied to each row $\boldsymbol{X}_{j,*}$ of the input matrix. The operation is defined as:

$$\boldsymbol{U}_{j,*} = \text{Drop}\left(\sigma\left(\boldsymbol{W}_2 \boldsymbol{X}_{j,*} + \boldsymbol{b}_2\right)\right),$$
$$\text{FM}_{C \to C}(\boldsymbol{X})_{j,*} = \text{Norm}\left(\boldsymbol{X}_{j,*} + \text{Drop}\left(\boldsymbol{W}_3 \boldsymbol{U}_{j,*} + \boldsymbol{b}_3\right)\right),$$
$$\forall j = 1, \dots, L,$$

where $\boldsymbol{W}_2, \boldsymbol{W}_3 \in \mathbb{R}^{C \times C}$ and $\boldsymbol{b}_2, \boldsymbol{b}_3 \in \mathbb{R}^C$.

When it is necessary to project the features to a different size $H$ ($H \neq C$), TSMixer applies a linear transformation to the residual term:

$$\text{FM}_{C \to H}(\boldsymbol{X})_{j,*} = \text{Norm}\left(\boldsymbol{W}_H \boldsymbol{X}_{j,*} + \boldsymbol{b}_H + \text{Drop}\left(\boldsymbol{W}_3 \boldsymbol{U}_{j,*} + \boldsymbol{b}_3\right)\right),$$
$$\forall j = 1, \dots, L,$$

where $\boldsymbol{W}_3, \boldsymbol{W}_H \in \mathbb{R}^{H \times C}, \boldsymbol{b}_3, \boldsymbol{b}_H \in \mathbb{R}^H$.

**Conditional Feature Mixing** The Conditional Feature Mixing (CFM) is a variation of the FM block that takes into account an associated static feature $\boldsymbol{S} \in \mathbb{R}^{1 \times C_s}$ in addition to the input sequence $\boldsymbol{X} \in \mathbb{R}^{L \times H}$. The block is designed to transform hidden features depending on the static features. The operation is defined as:

$$\boldsymbol{V}_{j,*} = \text{FR}_{C_s \to H}(\text{Expand}_L(\boldsymbol{S}))$$
$$\text{CFM}_{C \to H}(\boldsymbol{X}, \boldsymbol{S})_{j,*} = \text{FM}_{C+H \to H}(\boldsymbol{X} \oplus \boldsymbol{V})_{j,*} \tag{6}$$
$$\forall j = 1, \dots, L, \tag{7}$$

where $\text{Expand}_L(\cdot)$ expands the input along the time dimension by repeating it $L$ times, $\boldsymbol{V} \in \mathbb{R}^{L \times H}$ and $\boldsymbol{X} \oplus \boldsymbol{V} \in \mathbb{R}^{L \times (C+H)}$ is the concatenation of $\boldsymbol{X}$ and $\boldsymbol{V}$ along the feature dimension.

**Mixer Layer and Conditional Mixer Layer**  The Mixer Layer (Mix) is a composition of the Time Mixing and Feature Mixing, whereas the Conditional Mixer Layer (CMix) is a composition of the Time Mixing and Conditional Feature Mixing. Both Mix and CMix blocks apply the temporal and feature transformations respectively:

$$
\begin{aligned}
\text{Mix}_{C \to H}(\boldsymbol{X}) &= \text{FR}_{C \to H}\left(\text{TR}_{L \to L}(\boldsymbol{X})\right) \\
\text{CMix}_{C \to H}(\boldsymbol{X}, \boldsymbol{S}) &= \text{CFR}_{C \to H}\left(\text{TR}_{L \to L}(\boldsymbol{X}), \boldsymbol{S}\right).
\end{aligned}
$$

$$(8)$$

### B.3.2  Basic TSMixer for Multivariate Time Series Forecasting

For long-term time series forecasting (LTSF) tasks, TSMixer only uses the historical target time series $\boldsymbol{X}$ as input. A series of mixer blocks are applied to project the input data to a latent representation of size $C$. The final output is then projected to the prediction length $T$:

$$
\begin{aligned}
\boldsymbol{O}_1 &= \text{Mix}_{C \to C}(\boldsymbol{X}) \\
\boldsymbol{O}_k &= \text{Mix}_{C \to C}(\boldsymbol{O}_{k-1}), \forall k = 2, \ldots, K \\
\hat{\boldsymbol{Y}} &= \text{TP}_{L \to T}(\boldsymbol{O}_K)
\end{aligned}
$$

where $\boldsymbol{O}_k$ is the latent representation of the $k$-th mixer block and $\hat{\boldsymbol{Y}}$ is the prediction. We project the sequence to length $T$ after the mixer blocks as $T$ may be quite long in LTSF tasks. To increase the model capacity, we modify the hidden layers in Feature Mixing by using $\boldsymbol{W}_2 \in \mathbb{R}^{H \times C}, \boldsymbol{W}_3 \in \mathbb{R}^{C \times H}, \boldsymbol{b}_2 \in \mathbb{R}^H, \boldsymbol{b}_3 \in \mathbb{R}^C$ in Eq. equation B.3.1, where $H$ is a hyper-parameter indicating the hidden size. Another modification is using pre-normalization (Xiong et al., 2020) instead of post-normalization in residual blocks to keep the input scale.

### B.3.3  Extended TSMixer for Time Series Forecasting with Auxiliary Information

Given input data consisting of a target time series $\boldsymbol{X} \in \mathbb{R}^{L \times C}$, historical features $\hat{\boldsymbol{X}} \in \mathbb{R}^{L \times C_x}$, apriori known future features $\boldsymbol{Z} \in \mathbb{R}^{T \times C_z}$, and static features $\boldsymbol{S} \in \mathbb{R}^{1 \times C_s}$, TSMixer applies a series of conditional feature mixing and conditional mixer layers to project the input data to a latent representation of size $H$. The operation of a the architecture consisting $K$ blocks is defined as:

$$
\begin{aligned}
\boldsymbol{X}' &= \text{CFM}_{C+C_x \to H}(\text{TL}_{L \to T}(\boldsymbol{X} \oplus \hat{\boldsymbol{X}}), \boldsymbol{S}) \\
\boldsymbol{Z}' &= \text{CFM}_{C_z \to H}(\boldsymbol{Z}, \boldsymbol{S}) \\
\boldsymbol{O}_1 &= \text{CMix}_{2H \to H}(\boldsymbol{X}' \oplus \boldsymbol{Z}', \boldsymbol{S}) \\
\boldsymbol{O}_k &= \text{CMix}_{H \to H}(\boldsymbol{O}_{k-1}, \boldsymbol{S}), \forall k = 2, \ldots, K
\end{aligned}
$$

where $\boldsymbol{X}' \in \mathbb{R}^{T \times H}$ is the latent representation of all past information projected to the prediction length, $\boldsymbol{Z}' \in \mathbb{R}^{T \times H}$ is the latent representation of future features, $\boldsymbol{O}_k \in \mathbb{R}^{T \times H}$ is the output of the $k$-th mixer block. The final output, $\boldsymbol{O}_K$, is then linearly projected to the prediction space, which can be real values or the parameters of a probability distribution (e.g. negative binomial distribution that is commonly used for demand prediction (Salinas et al., 2020)).

## C  Experimental Setup

### C.1  Long-term time series forecasting datasets

For the long-term forecasting datasets (ETTm2, Weather, Electricity, and Traffic), we use publicly-available data that have been pre-processed by Wu et al. (2021), and we follow experimental settings used in recent papers (Liu et al., 2022b; Zhou et al., 2022a; Nie et al., 2023). Specifically, we standardize each covariate independently and do not re-scale the data when evaluating the performance. We train each model with a maximum 100 epochs and do early stopping if the validation loss is not improved after 5 epochs.

### C.2 M5 dataset

We obtain the M5 dataset from Kaggle[1]. Please refer to the participants guide to check the details about the competition and the dataset. We refer to the example script in GluonTS (Alexandrov et al., 2020)[2] and the repository of the third place solution[3] in the competition to implement our basic feature engineering. We list the features we used in our experiment in Table 7.

Table 7: Static features and time-varying features used in our experiments.

| Static features | Time-varying features |
|---|---|
| state_id | snap_CA |
| store_id | snap_TX |
| category_id | snap_WI |
| department_id | event_type_1 |
| item_id | event_type_2 |
| mean_sales | normalized_price_per_item |
| | normalized_price_per_group |
| | day_of_week |
| | day_of_month |
| | day_of_year |
| | sales (prediction target, only available in history) |

Our implementation is based on GluonTS. We use TFT and DeepAR provided in GluonTS, and implement PatchTST, FEDformer, and our TSMixer ourselves. We modified these models if necessary to optimize the negative binomial distribution, as suggested by DeepAR paper (Salinas et al., 2020). We train each model with a maximum 300 epochs and employ early stopping if the validation loss is not improved after 30 epochs. We noticed that optimizing other objective function might get significantly worse results when evaluate WRMSSE. To obtain more stable results, for all models, we take the top 8 hyperparameter settings based on validation WRMSSE and train them for an additional 4 trials (totaling 5 trials) and select the best hyperparameters based on their mean validation WRMSSE, then report the evaluation results on the test set. The hyperparameter settings can be found in Appendix E.

## D Effects of Lookback Window Size

We show the effects of different lookback window size $L = \{96, 336, 512, 720\}$ with the prediction length $T = \{96, 192, 336, 720\}$ on ETTm2, Weather, Electricity, and Traffic. The results are shown in Fig. 7.

## E Hyperparameters

Table 8: Hyerparamter tuning spaces and best configurations for TSMixer and TFT on long-term forecasting benchmarks.

| Search space | | ETTh1 | | | | |
|---|---|---|---|---|---|---|
| | | Learing rate | Blocks | Dropout | Hidden size | Heads |
| Model | $T$ | 0.001, 0.0001 | 1, 2, 4, 6, 8 | 0.1, 0.3, 0.5, 0.7, 0.9 | 8, 16, 32, 64 | 4, 8 |
| TSMixer | 96 | 0.0001 | 6 | 0.9 | 512 | relu |
| | 192 | 0.001 | 4 | 0.9 | 256 | relu |
| | 336 | 0.001 | 4 | 0.9 | 256 | relu |
| | 720 | 0.001 | 2 | 0.9 | 64 | relu |

---

[1] https://www.kaggle.com/competitions/m5-forecasting-accuracy/data
[2] https://github.com/awslabs/gluonts/blob/dev/examples/m5_gluonts_template.ipynb
[3] https://github.com/devmofl/M5_Accuracy_3rd

| | | | | | | |
|---|---|---|---|---|---|---|
| TFT | 96 | 0.001 | | 0.3 | | |
| | 192 | 0.001 | | 0.1 | | |
| | 336 | 0.001 | | 0.1 | | |
| | 720 | 0.001 | | 0.1 | | |
| | | | | ETTh2 | | |
| Search space | | Learing rate | Blocks | Dropout | Hidden size | Heads |
| Model | $T$ | 0.001, 0.0001 | 1, 2, 4, 6, 8 | 0.1, 0.3, 0.5, 0.7, 0.9 | 8, 16, 32, 64 | 4, 8 |
| TSMixer | 96 | 0.0001 | 4 | 0.9 | 8 | relu |
| | 192 | 0.001 | 1 | 0.9 | 8 | relu |
| | 336 | 0.0001 | 1 | 0.9 | 16 | relu |
| | 720 | 0.0001 | 2 | 0.9 | 64 | relu |
| TFT | 96 | 0.0001 | | 0.9 | | |
| | 192 | 0.001 | | 0.9 | | |
| | 336 | 0.001 | | 0.7 | | |
| | 720 | 0.001 | | 0.7 | | |
| | | | | ETTm1 | | |
| Search space | | Learing rate | Blocks | Dropout | Hidden size | Heads |
| Model | $T$ | 0.001, 0.0001 | 1, 2, 4, 6, 8 | 0.1, 0.3, 0.5, 0.7, 0.9 | 8, 16, 32, 64 | 4, 8 |
| TSMixer | 96 | 0.0001 | 6 | 0.9 | 16 | relu |
| | 192 | 0.0001 | 4 | 0.9 | 32 | relu |
| | 336 | 0.0001 | 4 | 0.9 | 64 | relu |
| | 720 | 0.0001 | 4 | 0.9 | 16 | relu |
| TFT | 96 | 0.001 | | 0.5 | | |
| | 192 | 0.001 | | 0.3 | | |
| | 336 | 0.001 | | 0.3 | | |
| | 720 | 0.001 | | 0.9 | | |
| | | | | ETTm2 | | |
| Search space | | Learing rate | Blocks | Dropout | Hidden size | Heads |
| Model | $T$ | 0.001, 0.0001 | 1, 2, 4, 6, 8 | 0.1, 0.3, 0.5, 0.7, 0.9 | 8, 16, 32, 64 | 4, 8 |
| TSMixer | 96 | 0.001 | 8 | 0.9 | 256 | relu |
| | 192 | 0.0001 | 1 | 0.9 | 256 | relu |
| | 336 | 0.0001 | 8 | 0.9 | 512 | relu |
| | 720 | 0.0001 | 8 | 0.1 | 256 | relu |
| TFT | 96 | 0.0001 | | 0.7 | 512 | |
| | 192 | 0.0001 | | 0.3 | 256 | |
| | 336 | 0.0001 | | 0.3 | 128 | |
| | 720 | 0.0001 | | 0.1 | 512 | |
| | | | | Weather | | |
| Search space | | Learing rate | Blocks | Dropout | Hidden size | Heads |
| Model | $T$ | 0.001, 0.0001 | 1, 2, 4, 6, 8 | 0.1, 0.3, 0.5, 0.7, 0.9 | 8, 16, 32, 64 | 4, 8 |
| TSMixer | 96 | 0.0001 | 4 | 0.3 | 64 | relu |
| | 192 | 0.0001 | 8 | 0.7 | 32 | relu |
| | 336 | 0.0001 | 2 | 0.7 | 8 | relu |
| | 720 | 0.0001 | 8 | 0.7 | 16 | relu |
| TFT | 96 | 0.001 | 2 | 0.9 | 64 | |
| | 192 | 0.001 | 1 | 0.1 | 32 | |
| | 336 | 0.001 | 1 | 0.1 | 32 | |
| | 720 | 0.001 | 2 | 0.7 | 64 | |
| | | | | Electricity | | |
| Search space | | Learing rate | Blocks | Dropout | Hidden size | Heads |
| Model | $T$ | 0.001, 0.0001 | 1, 2, 4, 6, 8 | 0.1, 0.3, 0.5, 0.7, 0.9 | 64, 128, 256, 512 | 4, 8 |
| | 96 | 0.0001 | 6 | 0.7 | 32 | relu |

TSMixer

| | $T$ | Learing rate | Blocks | Dropout | Hidden size | Heads |
|---|---|---|---|---|---|---|
| | 192 | 0.0001 | 8 | 0.7 | 16 | relu |
| | 336 | 0.0001 | 6 | 0.7 | 64 | relu |
| | 720 | 0.001 | 6 | 0.7 | 64 | relu |
| TFT | 96 | 0.0001 | 4 | 0.5 | 32 | |
| | 192 | 0.0001 | 6 | 0.9 | 8 | |
| | 336 | 0.0001 | 4 | 0.1 | 8 | |
| | 720 | 0.001 | 4 | 0.3 | 64 | |
| | | Traffic | | | | |
| Search space | | Learing rate | Blocks | Dropout | Hidden size | Heads |
| Model | $T$ | 0.001, 0.0001 | 1, 2, 4, 6, 8 | 0.1, 0.3, 0.5, 0.7, 0.9 | 64, 128, 256, 512 | 4, 8 |
| TSMixer | 96 | 0.0001 | 8 | 0.7 | 256 | relu |
| | 192 | 0.0001 | 8 | 0.7 | 256 | relu |
| | 336 | 0.0001 | 6 | 0.7 | 512 | relu |
| | 720 | 0.0001 | 2 | 0.9 | 256 | relu |
| TFT | 96 | 0.001 | 4 | 0.3 | 64 | |
| | 192 | 0.001 | 4 | 0.9 | 64 | |
| | 336 | 0.001 | 6 | 0.7 | 128 | |
| | 720 | 0.0001 | 8 | 0.1 | 256 | |

Table 9: Hyerparamter tuning spaces and best configurations for all models on M5.

| | M5 | | | | |
|---|---|---|---|---|---|
| Search space | Learing rate | Blocks | Dropout | Hidden size | Heads |
| Model | 0.001 | 1, 2, 3, 4 | 0, 0.05, 0.1, 0.3 | 64, 128, 256, 512 | 4, 8 |
| PatchTST | 0.001 | 2 | 0 | 64 | |
| Autoformer | 0.001 | 2 | 0 | 128 | |
| FEDformer | 0.001 | 1 | 0 | 256 | |
| DeepAR | 0.001 | 2 | 0.05 | 256 | |
| TFT | 0.001 | 1 | 0.05 | 64 | 4 |
| TSMixer | 0.001 | 2 | 0 | 64 | |

# F   Alternatives to MLPs

In Section 3, we discuss the advantages of linear models and their time-step-dependent characteristics. In addition to linear models and the proposed TSMixer, there are other architectures whose weights are time-step-dependent. In this section, we examine full MLP and Convolutional Neural Networks (CNNs), as alternatives to MLPs. The building block of full MLPs applies linear operations on the vectorized input with $T \times C$ dimensions and vectorized output with $L \times C$ dimensions. As CNNs, we consider a 1-D convolution layer followed by a linear layer.

The results of this evaluation, conducted on the ETTm2 and Weather datasets, are presented in Table 10. The results show that while full MLPs have the highest computation cost, they perform worse than both TSMixer and CNNs. On the other hand, the performance of CNNs is similar to TSMixer on the Weather dataset, but significantly worse on ETTm2, which is a more non-stationary dataset. Compared with TSMixer, the main difference is both full MLPs and CNNs mix time and feature information simultaneously in each linear operation, while TSMixer alternatively conduct either time or feature mixing. The alternative mixing allows TSMixer to use a large lookback window, which is favorable theoretically (Section 3) and empirically (previous ablation), but also keep a reasonable number of parameters, which leads to better generalization. On the other hand, the number of parameters of conventional MLPs and CNNs grow faster than TSMixer when increasing the window size $L$, which may suffer higher chance of overfitting than TSMixer.

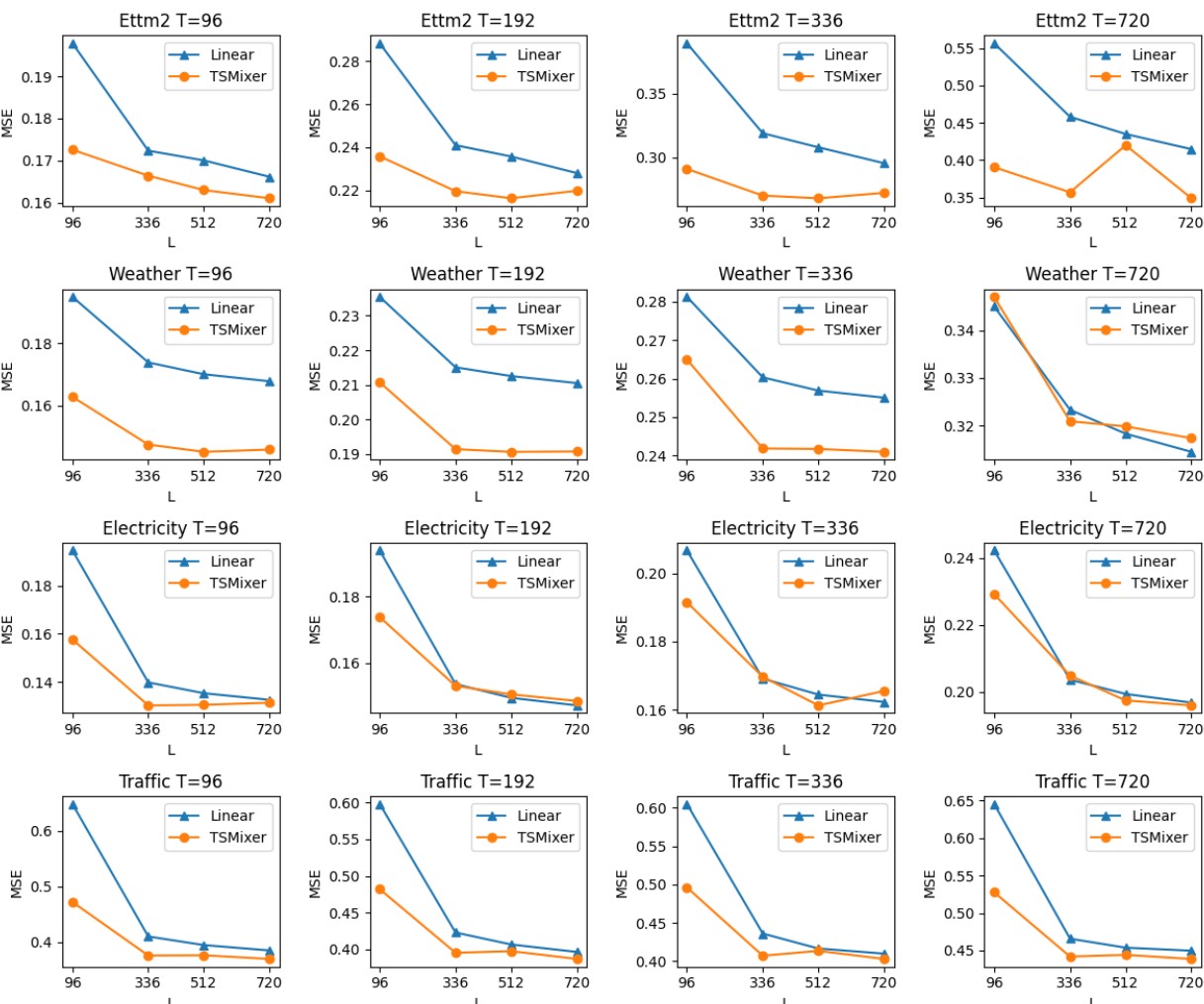

Figure 7: Effects of lookback window size on TSMixer.

Table 10: Comparison with other MLP-like alternatives.

| Models | | TSMixer | | Full MLP | | CNN | |
|--------|-----|-------|-------|-------|-------|-------|-------|
| Metric | | MSE | MAE | MSE | MAE | MSE | MAE |
| ETTm2 | 96 | **0.163** | **0.252** | 0.441 | 0.486 | 0.232 | 0.334 |
| | 192 | **0.216** | **0.290** | 1.028 | 0.755 | 0.323 | 0.410 |
| | 336 | **0.268** | **0.324** | 1.765 | 1.049 | 0.616 | 0.593 |
| | 720 | **0.420** | **0.422** | 2.724 | 1.305 | 2.009 | 1.214 |
| Weather | 96 | **0.145** | **0.198** | 0.190 | 0.279 | 0.149 | 0.220 |
| | 192 | **0.191** | **0.242** | 0.250 | 0.338 | 0.194 | 0.263 |
| | 336 | **0.242** | **0.280** | 0.298 | 0.375 | **0.242** | 0.306 |
| | 720 | 0.320 | **0.336** | 0.360 | 0.422 | **0.293** | 0.355 |

