# OpenReview forum: "TSMixer: An All-MLP Architecture for Time Series Forecast-ing"
_TMLR — Accepted by TMLR_

### Review · Reviewer_ZBio · 2023-05-31

**Summary Of Contributions:**

The paper proposes an MLP-based architecture for time series forecasting. The proposed approach consists of MLP layers across time as well as feature axis with residual connections and batch norm. The proposed approach achieves competitive performance or SOTA on several time series forecasting benchmarks.

**Audience:**

Yes

**Claims And Evidence:**

No

**Requested Changes:**

The paper should perform more experiments to understand why the proposed approach is not able to do well compared to univariate models on several benchmark datasets. The paper should also add more baselines to the M5 evaluations.

**Strengths And Weaknesses:**

### Strengths
- The paper focuses on an important problem of time series generation which can be useful in several practical use cases.
- The paper is easy to follow.

### Weaknesses
1. The proposed approach is lacking novelty as it is a direct application of MLP Mixer (Tolstikhin et al.) for time series forecasting.
2. While the proposed approach achieves SOTA on M5, it underperforms on several other time series forecasting benchmarks. Specifically, TSMixer is not able to outperform the linear models introduced in Zeng et al., 2023 on more than a few datasets.
3. As the paper points out, this is not the first work showing the effectiveness of linear models. Several recent works have shown this including Zeng et al., 2023,  Schiele et al., 2022.
4. The paper is missing some baselines on M5 datasets such as Autoformer.
5. Overall, the message of this paper seems unclear as it seems to only work on dataset with cross-variate interaction e.g. M5.



#### References
[1] Tolstikhin, Ilya O., et al. "Mlp-mixer: An all-mlp architecture for vision." Advances in neural information processing systems 34 (2021).
[2] Schiele, Philipp, Christoph Berninger, and David Rügamer. "ARMA Cell: A Modular and Effective Approach for Neural Autoregressive Modeling." arXiv preprint arXiv:2208.14919 (2022)

---

> ### Author Response · Authors · 2023-07-09
>
> We are grateful for your insightful comments and appreciate your time in providing feedback that helps enhance the quality of our work.
> 1. We understand your concern regarding the similarity between TSMixer and MLP-Mixer. While we acknowledge the structural resemblance, the conception of TSMixer primarily stemmed from our analysis of the unexpectedly strong performance of linear models. We have highlighted the differences between TSMixer and MLP-Mixer in Section 4.3. Moreover, the primary contribution of our work isn't confined to the model architecture alone. Our work elucidates why linear models effectively handle temporal patterns, substantiating previous findings of linear models outperforming existing multivariate models. We also point out the potential inadequacy of common benchmarks in evaluating multivariate models when cross-variate information proves irrelevant in those datasets. Perhaps most importantly for the field, we demonstrate that in practical applications, where cross-variate information is essential, multivariate models indeed supersede univariate models. Lastly, TSMixer sets a precedent as the first multivariate model that is comparable to univariate models on common benchmarks and establishes itself as the state-of-the-art model on real-world datasets. We appreciate your concern and we have modified the introduction section to better clarify the novelty of the paper.
> 2. While it's true that TSMixer didn't consistently outperform the linear models introduced by Zeng et al., 2023 on a few datasets, we underline that TSMixer has shown substantial improvement over other multivariate models. Our work presents TSMixer as the first multivariate model that stands in close competition with univariate ones, which may take advantage of inappropriate inductive biases from common benchmarks. We also emphasize that in most real-world applications, where cross-variate information is pivotal, multivariate models outperform univariate models, making TSMixer a robust choice across various scenarios. We have updated Table 3 to make these information clearer.
> 3. Our intention with this work is not to demonstrate the effectiveness of linear models. Instead, our investigations on linear models serve as a foundation to propose TSMixer and better understand why univariate models perform so well on current benchmarks and quantify their performance on more real-world datasets like M5.
> 4. We acknowledge your suggestion concerning the absence of Autoformer from our M5 dataset baselines. We opted to include FEDformer due to its superior performance over Autoformer on other benchmarks. We will include Autoformer results in M5 comparison soon.
> 5. As indicated in the paper, TSMixer is competitive with all univariate models and outperforms other multivariate models on the common benchmarks. The core message of our paper is to highlight the limitations of existing benchmarks and propose a model, TSMixer, that has simple architecture but performs optimally, irrespective of the usefulness of cross-variate information. We have updated the introduction section to improve the clarity of these messages.

---

> > ### Author Response · Authors · 2023-07-17
> >
> > We have added Autoformer results on M5 dataset. Please check Table 4, Table 6, and Table 9 for the updates. Thank you for the suggestion.

---

### Review · Reviewer_8WSL · 2023-05-31

**Summary Of Contributions:**

In this work, the authors introduce a novel MLP-based architecture (TSMixer) for time series forecasting. In particular, they stack multiple MLPs to capture temporal information and propose using MLPs in time-domain and feature-domain alternatively to leverage cross-variate information. Additionally, they propose feature alignment and mixing strategies to extend TSMixer to handle different types of  auxiliary features. The proposed architectures are evaluated on a variety of time-series datasets including seven multivariate long-term forecasting datasets and a large-scale challenging retail dataset M5. In their experiments, they demonstrate that their models perform comparably to SOTA univariate models and significantly outperform SOTA multivariate models. While achieving this performance, they show the proposed architectures have a much lower model size and faster inference speed, making them practical for real-world use.


**Audience:**

Yes

**Claims And Evidence:**

Yes

**Requested Changes:**

I would appreciate it if the authors could address the concerns in the Weaknesses, such as (1) and (4). Further clarification would be very helpful in solidifying my decision regarding the paper's acceptance.

**Strengths And Weaknesses:**

Strength:
- The authors propose a novel architecture (TSMixer) using only MLPs rather than RNNs or attention-based frameworks for time series forecasting. They further extend the proposed architecture to handle cross-variate information and auxiliary features for more complex tasks.
- The authors evaluate the proposed architectures on a variety of forecasting tasks, including multivariate long-term forecasting benchmarks and a large-scale retail forecasting task. The results demonstrate that the proposed models significantly outperform other SOTA multivariate models and perform comparably with SOTA univariate models.
- In terms of computational efficiency, the proposed architectures have much smaller model sizes compared to RNNs and Transformer-based models, and achieve faster inference speeds.

Overall, I think this paper is well-written, and easy to follow.

Weaknesses:
- The definition of N in Figures 1 and 3 is a bit confusing. It is defined as the number of time series in section 4.2. but it's confusing to me if the input metric should also include 'N' time series. Further clarification would be helpful.
- I recommend using different notation types to denote model layers or operation names to avoid confusion, especially in Figure 4.
- It would be better to add a row showing average values or other metrics to compare different data and models as a whole.
- I wonder if the authors have considered different fusion strategies for handling auxiliary features, such as training separate MLPs for future and static features, and then merging the prediction scores for final prediction (similar to late fusion).
- Minor issues: There are spacing errors on page 7 (section 4.1) and page 11 (section 5.2).

---

> ### Author Response · Authors · 2023-07-09
>
> We appreciate your valuable feedback and the effort you've invested in reviewing our work. Your comments have helped us improve the quality and clarity of our paper.
>
> 1. We apologize for the confusion regarding 'N' in Figures 1 and 3.  In these figures 'N' was meant to denote the number of layers, contrasting from its definition as the number of time series in Section 4.2. In the revised manuscript, we changed the number of time series to M and added descriptions to define ‘N’ in the caption of Fig. 1.
> 2. We appreciate the suggestion on distinct notation types in Figure 4. We have changed the notation of model layers in bold borders in the revised version. We hope it now better distinguishes the model layers and operations.
> 3. We concur that an average row in our tables would be beneficial for clearer overall comparison. We have added a new row indicating the overall improvement of TSMixer in the revision.
> 4. We sincerely appreciate your suggestion on more investigation of the fusion architecture. However, in order to maintain emphasis on our primary contributions, specifically the mixer architecture and the utilization of cross-variate information, we have chosen to leave further exploration of various fusion strategies for subsequent research. Your insightful suggestion, however, will undoubtedly serve as a critical point of consideration for our future work.
> 5. Finally, we thank you for pointing out the spacing errors in sections 4.1 and 5.2. We will carefully revise these sections to ensure proper formatting in the final manuscript.

---

### Review · Reviewer_BSK6 · 2023-07-04

**Summary Of Contributions:**

Given the unexpectedly optimal performance of temporal linear models on time series analysis in existing work, the authors attempted to analyse the effect of cross-variate information and auxiliary information for the time series forecastability and accordingly proposed TMix-Only and Time-Series Mixer (TSMixer) which stacks temporal linear models with nonlinearities and considers the cross-variate feedforward layers. Experimental results on common academic benchmark datasets and the large-scale challenging retail dataset also support the general effectiveness of TSMixers on time series analysis.

**Audience:**

Yes

**Broader Impact Concerns:**

There is no broader impact concerns from my perspective.

**Claims And Evidence:**

Yes

**Requested Changes:**

1. It would be clearer to also include mathematical formulae in Section 4.1 when introducing TSMixer.
2. More information can be provided on the training techniques for TSMixer. It would be easier to elaborate them if the program is released.

**Strengths And Weaknesses:**

Strengths:

1. This paper is well-presented with a clear intuition.
2. This work is of high importance and relevance to time series analysis. It is a smart attempt to stack temporal linear models for multivariate time series forecasting and that with auxiliary information, considering the scale of a dataset and the long-term dependency in datasets. It also further analyses the effect of cross-covariate information for time series forecasting in different scenarios.
3. It is a promising idea to apply the mixer to time series analysis, adopting the effective method from computer vision.

Weaknesses:

1. An important scenario occurs to time series is volatility. Sometimes the non-periodic part, i.e., g(t) may not be Lipschitz continuous. It is worth a discussion.
2. The novelty of TSMixer needs to be further articulated. What distinguishes TSMixer from past mixing methods in computer vision (Tolstikhin et al., 2021) should be further emphasized.

Reference

Ilya O. Tolstikhin, Neil Houlsby, Alexander Kolesnikov, Lucas Beyer, Xiaohua Zhai, Thomas Unterthiner, Jessica Yung, Andreas Steiner, Daniel Keysers, Jakob Uszkoreit, Mario Lucic, and Alexey Dosovitskiy. Mlp-mixer: An all-mlp architecture for vision. In Marc’Aurelio Ranzato, Alina Beygelzimer, Yann N. Dauphin, Percy Liang, and Jennifer Wortman Vaughan (eds.), *Advances in Neural Information Processing Systems 34: Annual Conference on Neural Information Processing Systems 2021, NeurIPS 2021, December 6-14, 2021, virtual*, pp. 24261–24272, 2021.

---

> ### Author Response · Authors · 2023-07-09
>
> We are appreciative of your valuable insights and comments which allow us to improve the clarity and impact of our work. We have modified our draft as described below.
>
> 1. We agree that volatility can be another important phenomenon for real-world time series data, making the patterns non-periodic and non-smooth. A key challenge in this scenario could be relying solely on past observed temporal information. As one of the key messages of our work, we highlight the importance of utilizing cross-variate information and how our proposed architecture is capable of doing so effectively. We have expanded our discussion in Section 3 to address this aspect more comprehensively.
> 2. We appreciate your question on TSMixer's distinctiveness. While it shares architectural similarities with MLP-Mixer, our development of TSMixer, motivated by our analysis in Section 3, led to a unique normalization approach. In TSMixer, two dimensions represent features and time steps, unlike MLP-Mixer's features and patches. Consequently, we apply 2D normalization to maintain scale across features and time steps, since we have discovered the importance of temporal patterns in forecasting. Besides, we have proposed an extended version of TSMixer to better utilize the heterogeneous inputs, which is essential to achieve state-of-the-art results in real-world scenarios. We have highlighted these differences in Section 4.
> 3. We appreciate your suggestion to include mathematical formulae for improved clarit.. Given space constraints, we've incorporated these mathematical details in the Appendix.
> 4. Thank you for the suggestion. Due to the space limit, some training details are described in the Appendix. Please let us know if there is anything missing. We also have included our program in the supplementary materials, where you can find the implementation of the original TSMixer at `tsmixer_basic/models/TSMixer.py` and the extended version at `tsmixer_extended/src/gluonts/torch/model/tsmixer/`. We also plan to make our code publicly available upon the publication of our paper, which will facilitate a deeper understanding of the training process.

---

### Author Response · Authors · 2023-07-09
**General Response to Reviewers**

Dear Reviewers,

We deeply appreciate the time and expertise you have invested in reviewing our work and providing insightful recommendations. In response to your valuable feedback, we have revised our manuscript and highlighted the modifications in red for your convenience.

Your constructive comments have significantly contributed to the improvement of our work. We continue to welcome any further suggestions you may have. We look forward to engaging in further discussions to enhance the quality of our research.

---

> ### Author Response · Authors · 2023-07-17
> **Looking forward to your reply**
>
> Dear Reviewers,
>
> The author-reviewer discussion period will be end soon.
> We hope that our responses have answered your questions and the modifications have addressed the concerns. If you have any other questions or suggestions, please let us know and we respond as soon as possible. Looking forward to further discussion.

---

### Decision · Action_Editors · 2023-09-02

**Recommendation:** Accept with minor revision

**Comment:**

This paper attempts to analyse the effect of cross-variate information and auxiliary information for the time series forecastability and accordingly proposed TMix-Only and Time-Series Mixer (TSMixer) which stacks temporal linear models with nonlinearities and considers the cross-variate feedforward layers. The authors evaluate the proposed architectures on a variety of forecasting tasks, including multivariate long-term forecasting benchmarks and a large-scale retail forecasting task. The results demonstrate that the proposed models significantly outperform other SOTA multivariate models and perform comparably with SOTA univariate models. In terms of computational efficiency, the proposed architectures have much smaller model sizes compared to RNNs and Transformer-based models, and achieve faster inference speeds. It is a promising idea to apply the mixer to time series analysis, adopting the effective method from computer vision.  During author-reviewer discussions, some concerns related to novelty and the comments on volatility have been addressed successfully by the authors. The difference with the previous work MLP-Mixer is also well explained.  One reviewer still has concerns that the paper is direct application of MLP-Mixer for time series forecasting, lacking in terms of both novelty and experimentation.  Considering the TMLR criteria, I recommend that the paper can be accepted with minor revision.  The authors are highly encouraged to consider all reviewers's comments carefully when preparing the revision. The following revisions are necessary.

- The comment on volatility are answered in Section 3 Limitations of the analysis in the revised manuscript. The authors promised to scrutinize it in the future work and also mentioned the possible importance of multivariate information for the analysis on volatility.
- The discussion on the novelty compared with the previous work MLP-Mixer needs to be added.
- The experimental result of Autoformer on M5 datasets should be added, as promised by authors.

**Audience:**

The paper focuses on an important problem of time series generation which can be useful in several practical use cases.

**Claims And Evidence:**

The authors propose a novel architecture (TSMixer) using only MLPs rather than RNNs or attention-based frameworks for time series forecasting. They further extend the proposed architecture to handle cross-variate information and auxiliary features for more complex tasks. Experimental results on common academic benchmark datasets and the large-scale challenging retail dataset also support the general effectiveness of TSMixers on time series analysis.

---

> ### Author Response · Authors · 2023-09-11
>
> We are deeply grateful to all the reviewers and AE for the constructive feedback that significantly contributed to the enhancement of our research. We have now submitted the camera-ready version with the following amendments:
>
> - Addressed the comments on volatility in the "Limitations of the Analysis" section. This can be found in the concluding remarks of Section 3.
> - Enhanced our discussion on the novelty of our work in comparison to MLP-Mixer. This has been incorporated in Section 4.3.
> - We have included the experimental results of Autoformer on the M5 datasets. These can be found in Table 4 and Table 6.
>
> Once again, thank you for your guidance throughout this process.

---

> > ### Comment · Action_Editors · 2023-09-11
> > **Link to the OpenReview page**
> >
> > Thank you for authors' efforts for revising the manuscript. The minor revisions are confirmed.
> >
> > According to TMLR guidance, the link to the OpenReview page is needed in the camera-ready version.

---

> > > ### Author Response · Authors · 2023-09-11
> > >
> > > Thanks for pointing out the problem.
> > > We've updated the submission. Please check.